# Investigation on the Influence of Production and Incubation Temperature on the Growth, Virulence, Germination, and Conidial Size of *Metarhizium brunneum* for Granule Development

**DOI:** 10.3390/jof9060668

**Published:** 2023-06-14

**Authors:** Tanja Seib, Katharina Fischer, Anna Maria Sturm, Dietrich Stephan

**Affiliations:** 1Julius Kühn-Institute, Federal Research Centre for Cultivated Plants, Institute for Biological Control, Schwabenheimerstraße 101, 69221 Dossenheim, Germany; tanja.seib@julius-kuehn.de (T.S.);; 2Technical University Darmstadt, Department Biologie, Schnittspahnstraße 4, 64287 Darmstadt, Germany

**Keywords:** conidial size, germination, granule, growth, *Metarhizium brunneum*, temperature, virulence

## Abstract

Important for the infection of an insect with an entomopathogenic fungus and its use as a plant protection agent are its growth, conidiation, germination, and virulence, which all depend on the environmental temperature. We investigated not only the effect of environmental temperature but also that of production temperature of the fungus. For this purpose, *Metarhizium brunneum* JKI-BI-1450 was produced and incubated at different temperatures, and the factors mentioned as well as conidial size were determined. The temperature at which the fungus was produced affects its subsequent growth and conidiation on granule formulation, the speed of germination, and the conidial width, but not its final germination or virulence. The growth and conidiation was at its highest when the fungus was produced at 25 °C, whereas when the germination was faster, the warmer the fungus was produced. The incubation temperature optimum of JKI-BI-1450 in relation to growth, speed of germination, and survival time was 25–30 °C and for conidiation 20–25 °C. Conidial length decreased with increasing incubation temperature. Although the fungus could not be adapted to unfavorable conditions by the production temperature, it was found that the quality of a biological control agent based on entomopathogenic fungi can be positively influenced by its production temperature.

## 1. Introduction

The aim of this study was to investigate the effects of different temperatures on the growth, conidiation, conidial size, germination and virulence of a *Metarhizium*-based soil granule, which was developed to control pest insects in the soil, especially in potato cultivation. Therefore, we developed a formulation technology for granules based on entomopathogenic fungi [1], and examined whether this technology can be used for *Metarhizium brunneum* strain JKI-BI-1450. To produce our granule, the fungus was initially produced in liquid medium. Subsequently, a thin layer of biomass was coated on autoclaved millet by fluid bed drying.

The advantage of applying a soil granule produced this way is that only after its application in the soil and the associated contact with moisture does the fungus overgrow the granule (Figure 1) and conidiate on the surface, which in turn leads to the pest insect being infected once contact has been established with the conidia [2]. The direct application of conidia into the soil has the advantage that the infectious unit of the fungus is directly present, however the product is dusting, and thus poses a risk for both the user and bystander.

Therefore, the growth and conidiation of the fungus on the granule in the soil are essential for its effectiveness, and for both of these processes, the temperature of the environment is important. Generally, temperatures between 20 to 30 °C are optimal for the growth and sporulation of entomopathogenic fungi, which is also valid for *Metarhizium* [3,4,5,6]. However, the temperature optimum varies between different *Metarhizium* species. Kryukov et al. [7] demonstrated that *M. brunneum* exhibited the best growth at 25 °C, whereas the optimum growth temperature of *Metarhizium robertsii* was 30 °C. Only *M. robertsii* grew at 35 or 37.5 °C, whereas *M. brunneum* was not able to grow at these temperatures. Keyser et al. [8] also showed that the examined *M. brunneum* species exhibited their maximum mycelial growth at 24 °C. Strains of *Metarhizium acridum*, *Metarhizium giuzhouense*, and *M. robertsii* had their maximum at 28 °C or even at 32 °C. 

One constraint in applying entomopathogenic fungi for controlling soil dwelling pests is the difference in soil temperatures and the temperature optimum of the fungi. The granule investigated in this study should later be used in cultivation at the same time when potato planting, where the soil temperature is lower than the optimum of the fungus [9]. An application at lower temperatures can lead to a reduced growth and conidiation of the fungus [6,10,11]. As conidia are the infection units of the fungus [12], lower levels of conidiation may lead to a reduction in efficacy [13]. We intended to take advantage of the fact that many microorganisms can cope better with stressful conditions when they have already been exposed to them [14,15,16,17]. Therefore, we investigated whether the fungus may be adapted to unfavorable temperatures by culturing it several times at these temperatures. Following growth and conidiation of the fungus on the granule, the next step to successful infection is the contact of the conidia with the insect cuticle. After adhesion on the cuticle, the conidia form a germ tube, which penetrates the cuticle through enzymatic and mechanical pressure. Moreover, germination is also heavily influenced by temperature [6,18]. Walstad et al. [19] and Dimbi et al. [3] demonstrated the highest germination for *Metarhizium* after 24 h at 25 °C. Similar results were determined by Hywel-Jones and Gillespie [20], with the fastest and highest germination having being observed after an incubation at 25 to 30 °C. Rath et al. [21] and Skalický et al. [22] showed that germination rates up to 100% were even possible at low temperatures, although such values are obtained slower compared to rates at higher temperatures. After adhesion, germination, and penetration, the fungus can multiply in the hemolymph and kill the insect either by nutrient deprivation or toxins, which some fungi are able to produce [23,24]. Apart from growth and germination, the virulence of entomopathogenic fungi is also temperature-dependent [6,25]. For *Metarhizium,* many studies showed the highest or fastest mortality at higher temperatures of 30 or 35 °C [3,26], or 25 to 30 °C [27,28], respectively.

Investigations on the influence of temperature on the growth, germination, and virulence of entomopathogenic fungi have often been described. The novelty of our study is that we examined both the influence of the production and incubation temperature. Initially, we investigated whether the production temperature of the biomass used for our granule-based formulation affects the growth of the fungus on this granule at different incubation temperatures. The purpose of these experiments was to verify whether the quality of the granule, and its ability to grow at sub-optimal temperatures could be improved by the previous production parameters of the biomass. In our application strategy, the infectious conidia were only formed after application in soil under moist conditions and at sufficient temperatures. Therefore, in the second step, we assessed whether the temperature for conidiation had an influence on their size, germination, and virulence to better understand the potential field conditions.

## 2. Materials and Methods

### 2.1. Fungal Strain

*Metarhizium brunneum* strain JKI-BI-1450 was isolated in 2013 at the Institute for Biological Control, Julius Kühn-Institute, Darmstadt (Germany) from an infected adult *Agriotes lineatus,* which was collected by Jörn Lehmhus in Braunschweig, Germany.

### 2.2. Maintenance of JKI-BI-1450

The fungal strain JKI-BI-1450 was stored at −80 °C in cryo-tubes (Microbank, Pro-Lab Diagnostics, Richmond Hill, Canada) and routinely cultured on malt peptone agar (MPA) containing 3% (*w*/*v*) malt extract (Merck, Darmstadt, Germany), 0.5% (*w*/*v*) peptone from soybean (Merck), and 1.8% (*w*/*v*) agar-agar (Roth, Karlsruhe, Germany).

### 2.3. Colony Size and Conidiation at Different Temperatures

JKI-BI-1450 was spread on MPA and incubated for 2 weeks at 25 °C in the dark. A total of 2 µL of a conidial suspension (1 × 10^8^ conidia/mL in sterile 0.5% (*v*/*v*) Tween 80^®^ (Merck)) of JKI-BI-1450 was pipetted in the center of the petri dishes filled with MPA. The dishes were then incubated for 14 days at 15, 20, 25, 30, and 37 °C in the dark, respectively. The diameter of the fungal colony was measured twice, perpendicular to one another. To determine the conidiation, 2 mL of sterile 0.5% (*v*/*v*) Tween 80^®^ was added to each dish and the conidia were scraped off with a sterile spatula. The conidia concentration was determined using a hemocytometer, and the conidia per mm^2^ was calculated based on the final colony size. The experiment was repeated six times independently with five replicates each.

### 2.4. Temperature Adaptation

JKI-BI-1450 was cultivated on MPA for 14 days at 15, 20, 25, and 30 °C in the dark, respectively. New MPA plates were then inoculated with the fungi produced at these temperatures and incubated for 14 days at the same temperatures as before in the dark. This subsequent cultivation was repeated three times. After that, the final adapted fungus cultures of each growth temperature were stored at −80 °C.

### 2.5. Effects of Production and Incubation Temperature on the Fungus Growth

The granules were produced as described by Stephan et al. [1]. For liquid culture inoculation, the temperature-adapted fungus cultures were taken from the −80 °C deep freezer and cultivated for 21 days on MPA at the corresponding temperature in the dark. For the liquid production of fungal biomass, 50 mL sterile medium containing 2.5% (*w*/*v*) glucose (Merck), 2% (*w*/*v*) corn steep (Sigma Aldrich, Buchs, Switzerland), and 0.5% (*w*/*v*) sodium chloride (Merck) in 100 mL flasks was used. Each flask was inoculated with 5 × 10^6^ conidia suspension of the previously described 21-day-old cultures in 1 mL of sterile 0.5% (*v*/*v*) Tween 80^®^ and were then incubated at 130 rpm on a horizontal shaker (Novotron, 50 mm deflection, Infors, Bottmingen, Switzerland). In liquid culture, the conidia of JKI-BI-1450 germinate and form mycelium and submerged spores. The duration of the production process is influenced by the production temperature. To determine the end of the exponential growth phase at each production temperature, the formation of the submerged spores was observed microscopically every 24 h. Based on these results, the incubation time at 15 °C was set to 120 h, and at 30 °C to 144 h, respectively. At 20 and 25 °C the fungus was cultivated for 72 h. The biomass suspension of each cultivation was centrifuged for 10 min at 25 °C and 15,344× *g.* The supernatant was then discarded, and the pellet was resuspended in 0.9% (*w*/*v*) sodium chloride solution and centrifuged again. This process was repeated twice. The dry weight of the biomass was determined using a moisture determination balance (Ma30, Sartorius, Göttingen, Germany), and a suspension of 3% dry biomass was prepared by diluting with a 0.9% (*w*/*v*) sodium chloride solution. From each biomass suspension a granule was prepared with a laboratory fluid-bed dryer (Strea-1, Aeromatic-Fielder AG, Bubendorf, Switzerland, nozzle diameter: 1 mm, Container volume: 16.5 L). In contrast to the process described by Stephan at al. [1], 100 g of autoclaved millet seeds were placed in the container, and 15 g of the 3% biomass suspension was sprayed by a top spray variant on the millet with a nozzle pressure of 2 bar. The granules produced in this way will be henceforth referred to as 15 °C-granule, 20 °C-granule, 25 °C-granule, and 30 °C-granule throughout this study. For each batch, ten granule grains (millet seeds covered with fungus) were placed on five petri dishes filled with agar (1% (*w*/*v*) agar-agar) and were then incubated at 15, 20, 25, and 30 °C for 14 and 28 days in the dark, respectively. After this incubation period, the number of colonized granule grains was determined. Granule grains covered with the mycelium or conidia of JKI-BI-1450 were counted as colonized. To determine the conidia concentration, 1 mL of sterile 0.5% (*v*/*v*) Tween 80^®^ was added to each petri dish. The granule grains and the suspension was transferred to a tube. The tubes were placed on a vortexer for 10 s and afterwards in an ultrasonic bath (Sonorex RK 52, 35 kHz, Bandelin electronic GmbH & Co. KG, Berlin, Germany) for 15 min. The conidia concentration was determined using a hemocytometer. Based on the colonization and the determined amount of conidia per plate, the conidia per granule grain were determined by dividing the amount of conidia by the number of colonized granule grains. This allowed to determine the conidiation independently of colonization.

Ten conidia of each petri dish were photographed, and the length and width was measured using the program cellSens Standard, Olympus (Figure 2). The conidial index was calculated by dividing the length by the width. The experiment was repeated independently three times.

### 2.6. Effect of More Unfavorable Incubation Temperature

A 25 °C-granule was produced and formulated as described. Ten granule grains were placed on five separate petri dishes filled with agar (1% (*w*/*v*) agar-agar) and were incubated at 5, 10, 25, and 35 °C for 14, 28, and 42 days in the dark, respectively. The granule colonization, the conidia per granule grain, and the conidial size were measured as described above. The experiment was repeated independently three times.

### 2.7. Effect of Simulated Soil Temperature

To simulate the soil temperature in the field during potato planting as realistically as possible, soil temperature data from a field trial in Uelzen Niedersachsen, Germany were used for adjusting the temperature profile of the incubator. The data were based on temperatures measured in 2018 and 2019, which were obtained from three points inside the potato mound. From these six measurements, averages were calculated every 4 h for 6 weeks, starting from the day of potato planting (7 May 2018 and 8 May 2019) [9]. A 25 °C-granule was produced as described above. Ten granule grains were placed on petri dishes filled with agar (1% (*w*/*v*) agar-agar). In total, 60 petri dishes were incubated at the simulated soil temperature, and an additional 60 petri dishes were incubated at a constant temperature of 25 °C in the dark. Ten petri dishes of each treatment were evaluated weekly over 6 weeks. The granule colonization, the conidia per granule grain, and the conidial size were measured as described above. The experiment was repeated independently three times.

### 2.8. Effect of Temperature on the Germination and the Virulence

#### 2.8.1. Preparation of Fungus Suspension

To determine the germination rate, the temperature-adapted fungus cultures were incubated for 21 days on MPA at the corresponding temperatures in the dark. An undefined number of conidia of each fungus was suspended in a 1.5 mL tube filled with 0.5% (*v*/*v*) Tween 80^®^. The tubes were placed on a vortexer for 10 s and afterwards in an ultrasonic bath (Sonorex RK 52, 35 kHz, Bandelin electronic GmbH & Co. KG) for 15 min. The conidia concentration was determined using a hemocytometer, and a conidial suspension of 1 × 10^6^ conidia/mL was made for each adapted fungus by diluting with 0.5% (*v/v*) Tween 80^®^. The length and width were determined for 100 conidia as previously described. 

#### 2.8.2. Speed of Germination

Twenty four droplets of each conidial suspension with a volume of 10 µL were placed on petri dishes filled with MPA, and were incubated at 15, 20, 25, and 30 °C in the dark, respectively. After 3, 6, 9, 12, 15, 18, 21, and 24 h, respectively, three droplets each were cut out of the agar and placed on a slide. On each agar piece, one hundred conidia were observed under the light microscope (×400), and the percentage of germinated conidia was subsequently determined. Conidia were rated as germinated when the germ tube was longer than the width of the conidia. The experiment was repeated independently three times.

#### 2.8.3. Final Germination Rate

To determine the final germination rate, the experiment was repeated as described above, except that 25 mg/L Benomyl (Sigma Aldrich) was added to the MPA. The germination rate was determined after incubating for 96 h at the corresponding temperatures in the dark. The experiment was repeated independently three times. 

#### 2.8.4. Effect of Temperature on the Virulence 

The temperature-adapted fungal cultures were incubated for 21 days on MPA at the corresponding temperatures in the dark. Five mL of 0.5% (*v*/*v*) Tween 80^®^ were added to each plate, following which the suspensions were filtered through four layers of gauze, the supernatant was sonicated (Sonorex RK 52, Bandelin electronic GmbH & Co. KG 35 kHz) for 15 min, and for each adapted fungal culture 5 mL of a conidial suspension with 1 × 10^7^ conidia/mL with 0.5% (*v*/*v*) Tween 80^®^ was prepared. Afterwards, *Galleria mellonella* larvae (larval stage 5–6) were dipped for 2–3 s into the four fungal suspensions, 0.5% (*v*/*v*) Tween 80^®^, or sterile deionized water. Each larvae was placed individually into a plastic box (7 cm diameter and 2.5 cm height). Ten larvae per treatment were incubated at 15, 20, 25, and 30 °C in the dark, respectively. The number of dead larvae was determined daily for a period of 14 days. The experiment was independently repeated four times. 

### 2.9. Statistical Analysis

The data was statistically analyzed with the software SAS Studio 3.8. Normality of the data was tested using the Shapiro–Wilk test and the homogeneity of variance was checked by the Levene’s test. To compare the colony size and the conidiation at different temperatures, and the effect of extreme incubation temperatures on the fungal growth, the Wilcoxon test (*p* < 0.05) was used. For analyzing the effects of production and incubation temperature, or incubation period on the fungal growth and the conidial size, a generalized linear model with Wald statistics for type 3 analysis and multiple comparison according to the Tukey test (GLMM, *p* < 0.05) was employed. For the analysis of the effect of one parameter (production temperature, incubation temperature, and incubation period), the data of the other two parameters were pooled. The effect of simulated soil temperature on the fungal growth was analyzed for each week by the non-parametric Mann–Whitney U-test (*p* < 0.05). Furthermore, the data of each incubation temperature was compared using a generalized linear model (*p* < 0.05). The final germination rate was determined to be above 99% in all treatments. Due to the method that was used, a more exact determination of the values was not possible, and therefore a statistical evaluation was not conducted. To compare the germination progress, the results of the germination speed and final germination rates were assembled, and τ was calculated for each treatment and every replicate. For the calculation of τ, a non-linear regression according to Dantigny et al. [29] (Formula (1)) was used. 

Formula (1)—Calculation of τ:(1)$$P=P_{max}[1-\frac{1}{1}+{\left(\frac{t}{\tau }\right)}^{d}]$$
where *P* is the percentage of germinated conidia in dependence of the maximum percentage of germination, (*P_max_*), *t* designates the germination time, τ is the point of time when 50% of the maximal germinated conidia have been germinated, and d is the design parameter. The design parameter was selected according to Dantigny et al. [29].

Furthermore, the slopes at the inflection point for each treatment and every replicate were calculated with Formula (2). 

Formula (2)—Calculation of the slope at the inflection point:(2)$$Slope=\frac{\left(d\times P_{max}\times q\right)}{{\tau \times q}^{\frac{1}{d}}\times {\left(q+1\right)}^{2}}$$

With $q=\left(\frac{d-1}{d+1}\right)$

For analyzing the effects of the production and incubation temperature on τ and the slope at the infection point a generalized linear model with Wald statistics for type 3 analysis and multiple comparison according to the Tukey test (GLMM, *p* < 0.05) was used. For the analysis of the effect of one parameter, the data of the other parameters were pooled.

To analyze the virulence, the mean survival time (ST_50_) for each replicate was calculated using a survival analysis (Kaplan–Meier) and compared using a generalized linear model (*p* < 0.05). 

For analyzing the effect of the production temperature and both controls, data of all incubation temperatures, whereas for the effect of the incubation temperature, data of all production temperatures were pooled. 

## 3. Results

### 3.1. Colony Size and Conidiation at Different Temperatures

The incubation temperature had a significant effect on the colony size (*x*^2^ = 137.3667; df = 4; *p* < 0.0001) and conidiation (*x*^2^ = 125.8045; df = 4; *p* < 0.0001). Moreover, the results demonstrate a significantly larger colony size of *M. brunneum* strain JKI-BI-1450 after incubation at 25 or 30 °C than after incubation at 15 or 20 °C (Figure 3). In contrast, the fungus formed significantly more conidia at 20 or 25 °C than at 15 or 30 °C, respectively. 

### 3.2. Effects of Production and Incubation Temperature on the Fungal Growth on the Soil Granule

The granule colonization by JKI-BI-1450 was found to have been significantly influenced by the production and incubation temperature of the biomass (production temperature: (*x*^2^ = 2298.94; df = 3; *p* < 0.001); and incubation temperature (*x*^2^ = 118.70; df = 3; *p* < 0.001)) (Table 1). In brief, over all incubation temperatures, granule colonization was 87% for the granules prepared with the biomass produced at 25 °C. At a production temperature of 20 and 30 °C, respectively, granule colonization was significantly reduced but still higher than 70%. When the biomass was produced at 15 °C, only 1% of granule grains were colonized. In contrast, when incubated at 15 °C across all production temperatures combined, maximum colonization was achieved at an average of 72%. With increasing incubation temperatures, the percentage of colonized granule grains decreased continuously, but did not reach less than 50%. The incubation period was found to exhibit no effects on the granule colonization (*x*^2^ = 0.27; df = 1; *p* = 0.6037).

Besides the granule colonization, the conidiation of JKI-BI-1450 on the granule grains was significantly influenced by the production temperature of the biomass (*x*^2^ = 432.21; df = 3; *p* < 0.0001). The granule-containing biomass produced at 25 °C formed the most conidia with 2.75 × 10^7^ conidia per granule grain, followed by the 20 °C-granule and 30 °C-granule over all incubation temperatures. The conidiation of the 15 °C-granule was a hundred times lower than the other granules and differed significantly from granules with the other production temperatures. In addition to the production temperature, the incubation temperature of the granules was found to significantly influence the conidiation (*x*^2^ = 285.84; df = 3; *p* < 0.0001). The conidia concentration on the granules incubated at 25 °C was significantly the highest with 3.17 × 10^7^ conidia per granule grain, followed by incubation at 20 °C. The conidiation was observed to be the lowest at 15 and 30 °C incubation. They differed significantly from the other incubation temperatures, but not from each other. The incubation period also significantly influenced the conidiation (*x*^2^ = 122.85; df = 1; *p* < 0.0001). The conidia per granule grain doubled from the second to the fourth week of incubation, and differed significantly from each other, summarized over all production and incubation temperatures.

In addition to colonization and conidiation, the conidial size was also examined. Since the fungal growth on the 15 °C-granule was extremely low, the conidial size could not be determined for this granule (Table 2). The production temperature of the biomass did not affect the length (*x*^2^ = 4.24; df = 2; *p* = 0.1199) but did significantly impact the width (*x*^2^ = 11.20; df = 2; *p* = 0.0037) and the conidial index (*x*^2^ = 13.21; df = 2; *p* = 0.0014). Significantly wider and more roundish conidia were formed on the granule-containing biomass produced at 20 °C compared to the higher temperatures. The incubation temperature of the granules also influenced the size of the conidia (length: (*x*^2^ = 542.28; df = 3; *p* < 0.0001); width: (*x*^2^ = 166.61; df = 3; *p* < 0.0001); and index; (*x*^2^ = 169.62; df = 3; *p* < 0.0001)). Conidia were significantly longer and wider when the granules were incubated at 15 °C compared to incubation at higher temperatures. The conidial index also revealed that the conidia were significantly more rounded when the granules were incubated at higher rather than lower temperatures. Furthermore, the conidia were significantly longer after an incubation period of 4 weeks than after 2 weeks (*x*^2^ = 11.24; df = 1; *p* = 0.0008). The incubation period was found to have no influence on the conidial width (*x*^2^ = 1.44; df = 1; *p* = 0.2306) or the conidial index (*x*^2^ = 1.86; df = 1; *p* = 0.1727).

### 3.3. Effect of More Unfavorable Incubation Temperature on the Fungal Growth

Within the investigation period, the fungus was not able to grow or to form conidia on the granule incubated at 5 or 35 °C (Table 3). The four incubation temperatures and three incubation periods resulted in significant differences in colonization (*x*^2^ = 173.0394; df = 11; *p* < 0.0001) and conidiation (*x*^2^ = 160.1709; df = 11; *p* < 0.0001). Incubating for 2 weeks at 10 °C revealed a significantly lower granule colonization, only reaching 80%, whereas 25 °C achieved nearly 100% granule colonization in the same timeframe. No significant differences were observed between 10 or 25 °C for incubation periods of 4 and 6 weeks, respectively. The conidiation was significantly lower after incubation at 10 °C rather than at 25 °C for each evaluation time point. After 2 weeks, JKI-BI-1450 did not form any conidia on the granule. After 4 weeks, approximately 5 × 10^3^ conidia per granule grain were formed. The conidia concentration increased a hundred-fold after incubating for an additional 2 weeks. When incubated at 25 °C, the conidia concentration was above 10^7^ at all evaluation time points.

In addition, significantly longer (*x*^2^ = 318.15; df = 1; *p* < 0.0001) and wider (*x*^2^ = 420.42; df = 1; *p* < 0.0001) conidia were formed on granules incubated at 10 °C compared to 25 °C (Table 4). The conidial index did not differ between these two incubation temperatures (*x*^2^ = 0.50; df = 1; *p* = 0.4796). The incubation period had no effect on the conidial size length: (*x*^2^ = 2.57; df = 1; *p* = 0.1089); width (*x*^2^ = 0.08; df = 1; *p* = 0.7716); and index (*x*^2^ = 2.54; df = 1; *p* = 0.1107)).

### 3.4. Effect of Simulated Soil Temperature on the Fungal Growth

First, the colonization at 25 °C was compared with the colonization at simulated soil temperature for each week independently. In this comparison of the incubation temperatures, the granule colonization was found to not differ significantly, except for the incubation periods of two and six weeks (Figure 4). Here, the colonization was significantly higher for the granules that were incubated at the simulated soil temperature compared to 25 °C (week 1: (*x*^2^ = 2.3677; df = 1; *p* = 0.1239); week 2: (*x*^2^ = 8.3617; df = 1; *p* = 0.0038); week 3: (*x*^2^ = 0.4229; df = 1; *p* = 0.5155); week 4: (*x*^2^ = 3.6004; df = 1; *p* = 0.0578); week 5: (*x*^2^ = 1.6223; df = 1; *p* = 0.2028); and week 6: (*x*^2^ = 50.7817; df = 1; *p* < 0.0001)). Secondly, the colonization of the fixed and simulated incubation temperature was compared separately over the experimental period. Both the incubation at 25 °C and at the simulated soil temperature showed no significant increases in colonization from the 2nd week onwards and was between 90 and 100% (25 °C: (*x*^2^ = 12.54; df = 5; *p* = 0.0281); and simulated soil temperature (*x*^2^ = 25.22; df = 5; *p* < 0.0001)).

Conidiation at 25 °C was also compared with the conidiation at simulated soil temperature for each week of the experiment. During week 4, the conidia concentration was significantly lower for the granules incubated at the simulated soil temperature rather than at 25 °C (week 1: (*x*^2^ = 17.1323; df = 1; *p* < 0.0001); week 2: (*x*^2^ = 15.5149; df = 1; *p* < 0.0001); week 3: (*x*^2^ = 13.6696; df = 1; *p* = 0.0002); week 4: (*x*^2^ = 1.3300; df = 1; *p* = 0.2488); week 5: (*x*^2^ = 0.2310; df = 1; *p* = 0.6308); and week 6: (*x*^2^ = 3.1751; df = 1; *p* = 0.0748)). Thereafter, the conidia per granule grain no longer differed. The conidiation at one incubation temperature was also compared separately over the duration of the experiment. During incubation at 25 °C the maximum conidia concentration was achieved after 2 weeks with 1.86–2.17 × 10^7^ conidia per granule grain (*x*^2^ = 88.47; df = 5; *p* < 0.0001), while during incubation at simulated soil temperature this was not achieved until after 4 weeks with 1.44–1.79 × 10^7^ conidia per granule grain (*x*^2^ = 197.58; df = 5; *p* < 0.0001).

Since no conidia were formed after 1 week of incubation at the simulating soil temperature, this time point was not considered when determining the conidial size. The granule formed significantly longer (*x*^2^ = 862.81; df = 1; *p* < 0.0001) and wider (*x*^2^ = 7.45; df = 1; *p* = 0.0063) conidia during incubation at the simulated soil temperature rather than at 25 °C, where the granule was significantly rounder (*x*^2^ = 285.16; df = 1; *p* < 0.0001) (Table 5). The incubation period also influenced the conidial size. After 3 weeks of incubation, the conidia were significantly the longest (*x*^2^ = 14.83; df = 4; *p* = 0.0051), whereas the conidia were the widest (*x*^2^ = 19.60; df = 4; *p* = 0.0006) after 4 weeks of incubation. In addition, the conidia were significantly most elongated after 2 and 3 weeks of incubation and became more round with increasing incubation time (*x*^2^ = 23.93; df = 4; *p* < 0.0001).

### 3.5. Effect of Temperature on Conidial Size and Germination

Incubation at 15 and 20 °C resulted in significantly longer (*x*^2^ = 234.64; df = 3; *p* < 0.0001) and rounder (*x*^2^ = 134.10; df = 3; *p* < 0.0001) conidia compared to higher temperatures (Table 6). However, the incubation temperature had no significant influence on the conidia width (*x*^2^ = 6.53; df = 3; *p* = 0.0887).

The germination was found to be faster when the fungus was produced at higher temperatures (Figure 5). This effect was observed across all incubation temperatures.

The statistical evaluation of this experiment is presented in Table 7. τ was reached significantly fastest with a production temperature of 30 °C (after 12.3 h), closely followed by 25 °C (after 12.8 h) (*x*^2^ = 1376.52; df = 3; *p* < 0.0001). When the fungus was produced at 15 and 20 °C, τ was reached significantly slower (3 h later). The incubation temperature also exhibited a significant influence on τ (*x*^2^ = 2822.81; df = 3; *p* < 0.0001). The higher the temperatures at which the fungus was incubated, the faster τ was reached. Τ achieved values between 11 h at 30 °C and 17 h at 15 °C, with a significant difference observed among all incubation temperatures.

The slope at the inflection point was also significantly influenced by the production temperature (*x*^2^ = 11.99; df = 3; *p* = 0.0074). When the fungus was produced at 20 °C, the slope was at its steepest, while at 25 °C, the slope was found to be at its lowest (Table 7). The slopes of the other two production temperatures were deemed to be in between. When incubated at the higher temperatures (25 or 30 °C, respectively), the slope of the germination rate was significantly steeper than when incubated at the lower temperatures (15 or 20 °C, respectively) (*x*^2^ = 92.95; df = 3; *p* < 0.0001).

### 3.6. Effect of Temperature on the Virulence

The production temperature of the fungus was found to have no influence on the ST_50_ but did differ significantly from the controls (*x*^2^ = 129.33; df = 5; *p* < 0.0001) (Table 8). The ST_50_ of the water control was significantly the highest with 10.83 days followed by the Tween 80^®^ control (7.63 days), and the fungi treatments with an average of 4 days. However, it was found that the survival time significantly decreased with increasing incubation temperature (*x*^2^ = 102.65; df = 3; *p* < 0.0001). ST_50_ was highest when the larvae were incubated at 15 °C, with an average of 6.03 days, and lowest at 30 °C, with an average of 3.16 days, respectively. The final mortality of the fungal treatments in relation to the production temperature did not differ significantly from each other (95–100%) but from the control treatments (*x*^2^ = 560.82; df = 5; *p* < 0.0001). The final mortality at 15 to 25 °C incubation was approximately 100% and differed significantly from the one observed at 30 °C (*x*^2^ = 17.17; df = 3; *p* = 0.0007).

## 4. Discussion

The *M. brunneum* strain JKI-BI-1450 can form mycelium and conidia across a wide temperature range. This fungus generated the largest colony size and the most conidia/mm^2^ at 25 °C. Similar results for the radial growth of *M. brunneum* were reported by Keyser et al. [8] and Kryukov et al. [7], respectively. More specifically, the optimum for mycelial growth of JKI-BI-1450 was 25–30 °C, and for conidiation 20–25 °C, respectively. Thomas and Jenkins [30], and Onsongo et al. [27], determined a higher temperature optimum for mycelia growth than for conidia formation for *Metarhizium anisopliae* and *Metarhizium flavoviride* as well. This effect has also been observed for other fungi such as *Mycosphaerella* var. *difformis fijiensis* [31,32], and *Pyrenophora semeniperda* [32]. These results suggest that the optima for mycelial growth and conidiation are slightly different. This could indicate that the fungus does not form mycelium and conidia uniformly in the field at fluctuating temperatures (day and night rhythm). As the soil temperature at the time of potato planting in Germany with 9.6–14.7 °C [9] is far below the optimum temperature of JKI-BI-1450 (25 °C), we tried to improve its growth under sub-optimal conditions. However, the assumption that the growth of the fungus at unfavorable temperatures can be improved by prior exposure to this stress could not be confirmed. We found no indications that the strain JKI-BI-1450 is able to adapt to unfavorable temperatures. Andrade-Linares et al. [17] also showed that not all of the fungi they assessed were able to increase their growth through priming. Most studies of stress adaptation have been conducted with conidia exposed to the stressor immediately prior to incubation. Therefore, further research on the quality of conidia of JKI-BI-1450 directly exposed to lower temperatures would be essential to clarify whether this fungus is able to adapt to stressful conditions. The results of this experiment show that the growth of the fungus and conidiation on the granules, which is crucial for its effectiveness, are dependent on and can be positively influenced by its previous production temperature. When the biomass, which was produced at the optimal growing temperature of 25 °C, was used for the formulation of the granule, both growth parameters were at their highest regardless of the incubation temperature. Therefore, the fungus is better able to grow at unfavorable temperatures on granules produced at this temperature than on ones produced at higher or lower temperatures.

The temperature during liquid production has an influence on the production of secondary metabolites of *Nigrospora* sp., which was previously shown by Arumugam et al. [33]. Furthermore, Kim et al. [34] demonstrated that the incubation temperature at liquid production causes the synthesis of phytases from *Aspergillus* sp. Therefore, it would be interesting to examine in further experiments whether the production temperature can affect the metabolism of JKI-BI-1450, which might have an impact on the growth of the granules. The production temperature can also influence further formulation steps. Jin et al. [35] showed that submerged spores of *Trichoderma harzianum* were able to survive vacuum drying better when produced at 32 °C, rather than at higher or lower temperatures. A varying survival of the biomass during fluid bed-drying could be a possible explanation for our diverse results of the different production temperatures. Another possible explanation could be that the different production temperatures resulted in a varying proportion of mycelia and submerged spores in the suspensions used for the production of the granules. The granules were prepared with a defined biomass concentration, not with a defined spore concentration. Stephan et al. [1] demonstrated that the mycelium, submerged spores, or conidia of *M. brunneum* JKI-BI-1339 were suitable for the production of a granule by fluid bed-drying. When we produced granules only based on mycelium of JKI-BI-1450, the fungus was not able to grow on this granule (Seib, unpublished data). Therefore, further experiments have to be conducted to determine the importance of the mycelium/spore proportion on the granule quality. In our experiments, the submerged spore concentration of the biomass suspensions were determined but the difference in the growth of the fungus on the different granules cannot be due to the amount of submerged spores, but maybe to a difference in the characteristics of the spores themselves. The spore concentration of the biomass suspensions averaged 3.7 × 10^6^ spores/mL.

Our results show the highest granule colonization after an incubation at 15 °C. However, this result is in contrast to both our and other results from different studies where a temperature optimum of *Metarhizium* was observed between 25 to 30 °C [3,5,6,19,20,36,37,38], respectively. We attribute this to an increased contamination at higher incubation temperatures. At elevated temperatures, the granules were more often colonized by bacteria than after incubation at lower temperatures. This prevented the growth of the fungus as a consequence. During the process of fluid bed-drying, non-sterile air is blown into the cylinder, which could lead to these contaminations. In subsequent experiments, the effects of higher colonization by incubation at lower temperatures was no longer observed. Therefore, and as the germination and virulence tests were performed exclusively with conidia and not with the granules, the results of all the following experiments were not influenced. The highest conidiation was found when incubated at 25 °C, which was also supported by the other experiments we conducted in this study. Thomas and Jenkins [30], and Tefera and Pringle [39], also showed a temperature optimum for conidiation at 24–25 °C for *Metarhizium*. On a 25 °C-granule, JKI-BI-1450 did not grow between 10 and 30 °C. The comparison of growth at 10 and 25 °C showed that after 4 weeks of incubation, there were no significant differences observed concerning granule colonization, whereas the conidia concentration at all evaluation times was lower at 10 °C than at 25 °C, respectively. The mycelium of *Metarhizium* strains can be formed at 10 °C with no or reduced sporulation, as previously confirmed by Skalický et al. [22]. To better understand the growth of the fungus under field conditions, a 25 °C-granule was incubated at a simulated soil temperature as well as at 25 °C, and the growth between the two was compared. Colonization did not differ, but conidiation was slower from the simulated soil temperature. The fungus formed conidia on the granule only after two weeks at simulated soil temperature. Consecutively, the pest insects in the soil can only become infected two weeks after an application of the granule. Since the main aim was to protect the progeny tubers and not the seed potato, a later conidiation would not necessarily present a problem. In fact, it would allow other micro- and macro-organisms in the soil time to colonize or feed on the grains.

An influence of fluctuating temperatures on the colony growth and virulence of *Beauveria bassiana* was investigated by Ghazanfar et al. [40]. The more the temperatures fluctuate, the more the growth was reduced compared to the growth at a constant 25 °C. Furthermore, the effect against *Heliothis virescens* was reduced by incubation at fluctuating temperatures. This effect was not seen for *Spodoptera littoralis*, which was also investigated.

The influence of the incubation temperature on the size of the conidia has been proven across all experiments. For strain JKI-BI-1450, incubation at lower temperatures always led to the formation of longer and partly wider conidia. Similar results were also shown by Glare et al. [41] for 16 *Metarhizium* strains from seven species. For other fungi, this effect were reported by Phillips [42,43] for *Monilinia fructicola*, by Tian and Bertolini [44] for *Monilinia laxa,* and by Tian and Bertolini [45] for *Botryfis allii* and *Penicillium hirsutum*, respectively. In contrast, Maitlo et al. [46] described that *Fusarium oxysporum* f. sp. *ciceris* formed the longest and broadest conidia when incubated at 30 °C, while at lower or higher temperatures the conidia were smaller. This strongly suggests that the relationship between the temperature and conidial size differs among the fungi of different genera and must be examined individually for each fungus. Further research must be conducted to understand the meaning of conidial size for biological activity. Since on the granule, infectious conidia were firstly formed in the soil, we investigated the effects of conidia production temperature, and thus their size, as well as the subsequent incubation temperature on germination and virulence.

We determined in our study that the higher the temperature at which the fungus was produced and incubated, the faster it reached the time at which 50% of the finally germinated conidia were germinated (τ). Our results are in contrast to Phillips [42], where conidia produced at lower temperatures germinated faster. This suggests that the relationship between the temperature and germination is strongly dependent on the fungus, as well as the conidial size. However, our experiments revealed a germination rate of approximately 90% after 24 h, and over 99% after 96 h at all production and incubation temperatures. At lower temperatures, the conidia germinated more slowly, but the amount of germinated conidia after 96 h was not lower than at higher temperatures. A comparable time delay of germination was also shown by Skalický et al. [22] for nine *M. anisopliae* strains comparing the germination rates at 15 and 20 °C after 24 and 48 h, respectively. Dimbi et al. [3] determined a much greater influence of the incubation temperature on the germination rate after 24 h for several *M. anisopliae* strains than within the results of our experiments. All strains showed a maximum germination of about 90% at 25 °C, whereas the germination rate at 15 °C was below 10%, respectively. Results on the influence of the germination speed on the virulence are inconsistent within the literature. Some studies show that rapidly germinating spores are more virulent [47,48,49,50,51], but there are also studies that conclude the opposite [52].

To investigate this further, the influence of the temperature on the virulence was examined. The production temperature, and thus the conidial size and the speed of germination had no effect on the ST_50_ or the final mortality of *G. mellonella*. This contradicts the results from Altre et al. [48] on the virulence of different strains of *Paecilomyces fumosoroseus* against *Plutella xylostella*, where larger conidia were more virulent. Furthermore, since the production temperature of the conidia affects their germination speed but not the final germination and virulence, it appears that a successful infection in this context depends on more than just fast germination. Our experiments showed that the higher the incubation temperature, the lower the ST_50_. In case of the fungus JKI-BI-1450, a high incubation temperature ranging from 25–30 °C was found to be correlated with a great mycelia growth in regard to colony size, fast germination, and fast virulence. Besides a fast germination, a faster growth of the fungus on the cuticle promotes a successful and faster penetration through the fungus by reaching the susceptible areas faster.

Dimi et al. [3], Bugeme et al. [26], Fargues et al. [28], and Onsongo et al. [27] demonstrated the highest or among the highest mortality of *Metarhizium* when incubated at 30 °C. In contrast, our results indicated a significantly lower mortality at 30 °C compared to the other incubation temperatures, but with a low ST_50_. The influence of higher incubation temperatures on the speed of insect development could be a possible explanation for this, as earlier molting potentially strips off conidia. To verify this, one would need to repeat the experiment with younger larvae.

In summary, we were able to show that the growth of an entomopathogenic fungus on a soil granule can be influenced and improved by its production conditions. Colonization and conidiation of a granule were at their highest when the fungus was produced at its optimum temperature. Furthermore, it was observed that the fungus was also able to grow on the granules and form conidia at lower temperatures and at simulated soil temperatures at the time of potato planting. This was particularly important, as our fungal strain and formulation process were selected for the control of wireworms in potatoes. However, the growth of the fungus, along with the germination and duration until lethal effect on the insects were found to be prolonged by lower temperatures. If this effect endangers the potential of the granule to control wireworms, an application at higher temperatures, for example in the year before planting the potatoes, would therefore give the fungus time to grow, sporulate, and establish in the soil, and reduce the population of the pest insect before potato cultivation.

## Figures and Tables

**Figure 1 jof-09-00668-f001:**
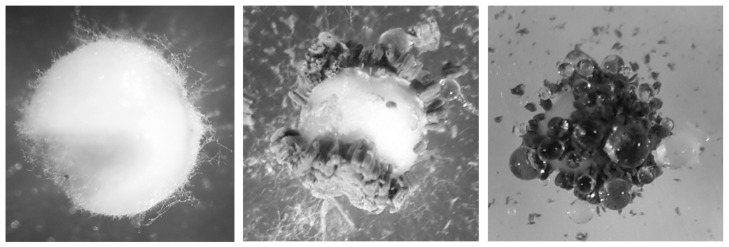
Growth and conidiation of *Metarhizium brunneum* JKI-BI-1450 on a granule grain.

**Figure 2 jof-09-00668-f002:**
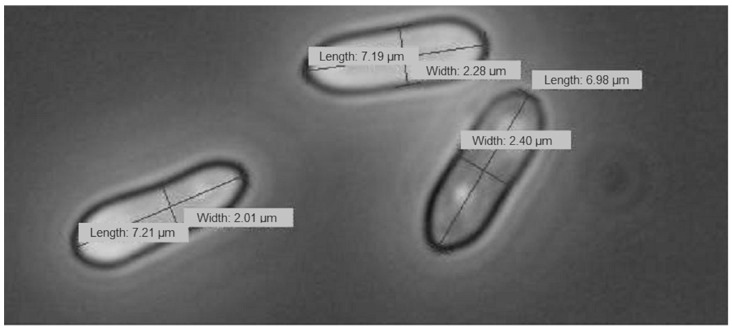
Measurement of conidia.

**Figure 3 jof-09-00668-f003:**
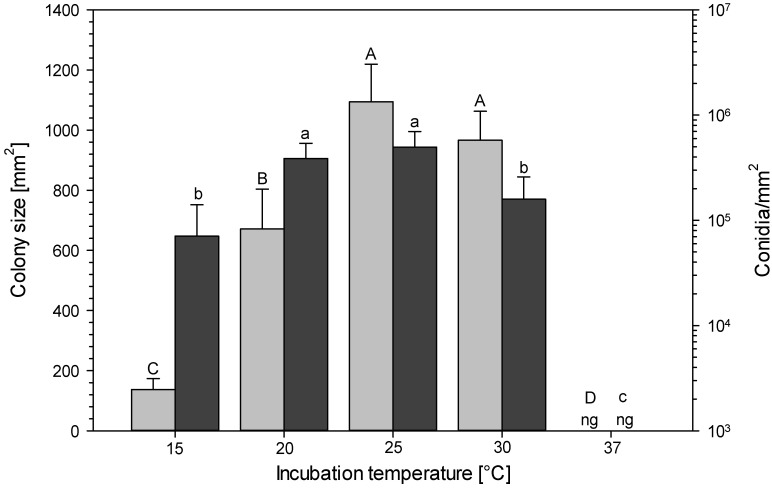
Influence of temperature on colony size (light grey) and conidia per mm^2^ (dark grey) of JKI-BI-1450 after 14 days (means and standard deviation). Different capital letters and corresponding lowercase letters indicate significant differences (Wilcoxon test, *p* < 0.05, *n* = 30), ng = no growth.

**Figure 4 jof-09-00668-f004:**
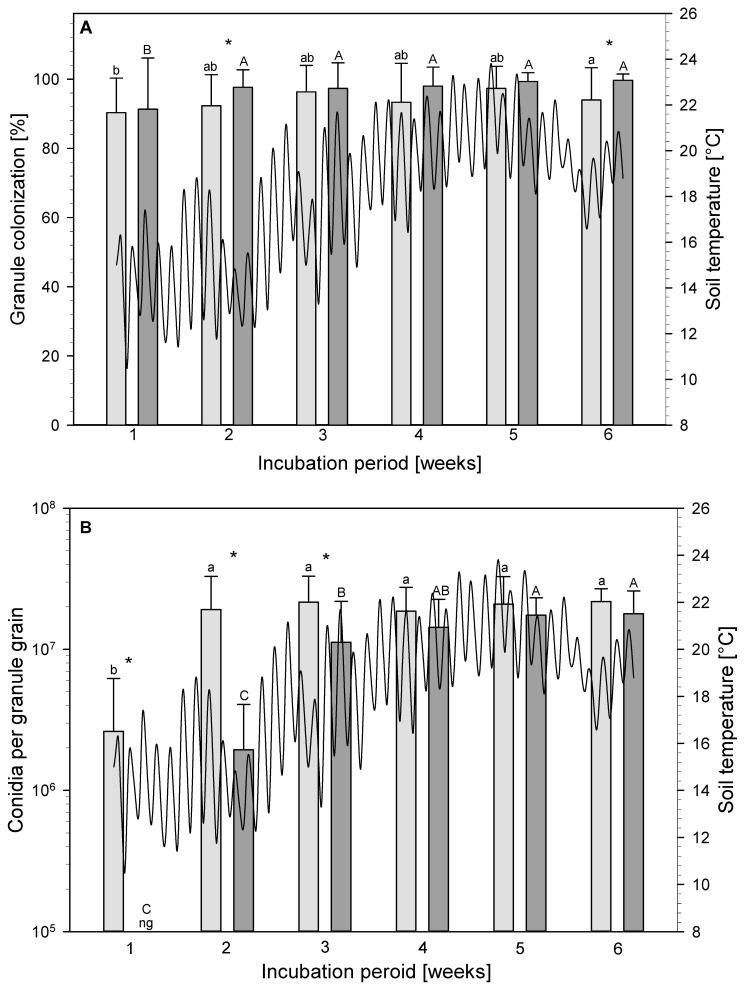
Effect of incubation at 25 °C (light grey) and at simulated soil temperature (dark grey) on the granule colonization (**A**), and on the number of conidia per granule grain (**B**) (means and SD). Means with * at one evaluation time are significantly different (Mann–Whitney U-Test, *p* < 0.5, *n* = 15). The differences between the evaluation times of the incubation at 25 °C are represented by different lower case letters and for the simulated soil temperature by different upper case letters (GLMM, *p* < 0.05, *n* = 30). The black line indicates the simulated soil temperatures from the day the potatoes were planted until 6 weeks thereafter (*n* = 3), ng = no growth.

**Figure 5 jof-09-00668-f005:**
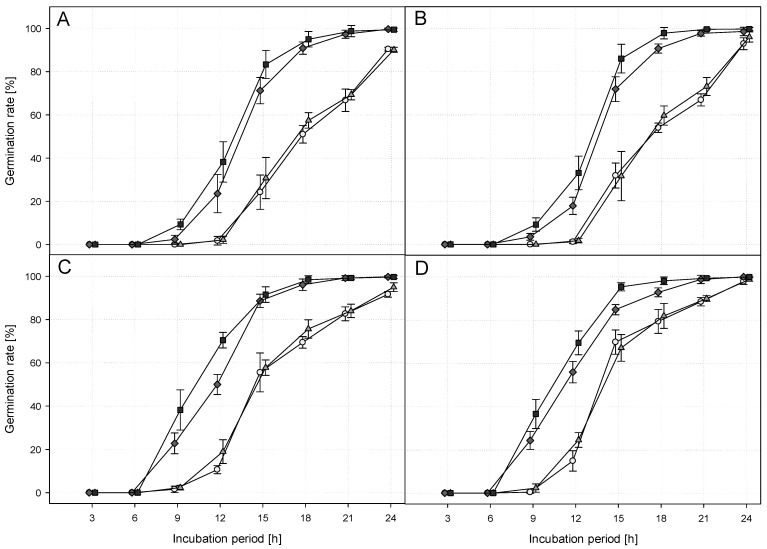
Effects of the production and incubation temperature on the germination rate of *Metarhizium brunneum* JKI-BI-1450 are shown (means and SD). The symbols show the different production temperatures (
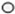
 = 15 °C, 
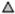
 = 20 °C, 
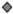
 = 25 °C, 
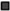
 = 30 °C) and the letters correspond to the incubation temperature ((**A**) = 15 °C, (**B**) = 20 °C, (**C**) = 25 °C, and (**D**) = 30 °C). The symbols are slightly offset for better visibility. *n* = 9.

**Table 1 jof-09-00668-t001:** Effects of the production and incubation conditions on the fungal growth of granules coated with the *Metarhizium brunneum* strain JKI-BI-1450.

Conditions		Growth Parameter				
		Granule Colonization [%]		Conidia per Granule Grain		*n*
Production temperature [°C]	15	1.66 (±4.34) *	D **	3.85 (±25.2) × 10^5^	C	120
	20	81.25 (±19.98)	B	2.47 (±1.70) × 10^7^	AB	
	25	87.17 (±17.40)	A	2.77 (±1.66) × 10^7^	A	
	30	72.25 (±23.24)	C	2.31 (±2.00) × 10^7^	B	
Incubation temperature [°C]	15	72.41 (±42.21)	A	1.09 (±1.18) × 10^7^	C	120
	20	61.58 (±37.28)	B	2.26 (±1.74) × 10^7^	B	
	25	56.92 (±36.53)	B	3.17 (±2.45) × 10^7^	A	
	30	50.92 (±36.69)	C	1.06 (±0.97) × 10^7^	C	
Incubation period [weeks]	2	60.83 (±39.15)	A	1.32 (±1.32) × 10^7^	B	240
	4	60.08 (±38.80)	A	2.47 (±2.19) × 10^7^	A	

* Mean ± standard deviation. ** Means of one condition at one growth parameter with the same letters are not significantly different. (GLMM, *p* < 0.05).

**Table 2 jof-09-00668-t002:** Effects of the production and incubation conditions on the conidial size of the *Metarhizium brunneum* strain JKI-BI-1450 formed on the granules.

Conditions		Conidial Size						
		Conidial Length [µm]		Conidial Width [µm]		Conidial Index *		*n*
Production temperature [°C]	20	6.70 (±0.58) **	A ***	2.21 (±0.22)	A	3.05 (±0.39)	B	1200
	25	6.75 (±0.58)	A	2.19 (±0.20)	B	3.10 (±0.38)	A	1200
	30	6.73 (±0.67)	A	2.19 (±0.22)	B	3.11 (±0.43)	A	1166
Incubation temperature [°C]	15	7.06 (±0.55)	A	2.26 (±0.22)	A	3.16 (±0.42)	A	900
	20	6.78 (±0.50)	B	2.15 (±0.19)	C	3.18 (±0.39)	A	900
	25	6.51 (±0.52)	C	2.16 (±0.19)	C	3.05 (±0.38)	B	900
	30	6.54 (±0.70)	C	2.22 (±0.23)	B	2.97 (±0.37)	C	866
Incubation period [weeks]	2	6.69 (±0.59)	B	2.19 (±0.21)	A	3.08 (±0.39)	A	1780
	4	6.76 (±0.63)	A	2.20 (±0.22)	A	3.10 (±0.41)	A	1786

* The index is calculated by dividing the length by the width. ** Mean ± standard deviation. *** Means of one condition at one conidial size parameter with the same letters are not significantly different. (GLMM, *p* < 0.05).

**Table 3 jof-09-00668-t003:** Effects of the incubation temperature and period on the fungal growth of a granule based on biomass produced at 25 °C.

Incubation Temperature [°C]	Incubation Period [weeks]	Growth Parameter				
		Granule Colonization [%] *		Conidia per Granule Grain		*n*
5	2	0 (±0)	C **	0 (±0)	C	15
	4	0 (±0)	C	0 (±0)	C	15
	6	0 (±0)	C	0 (±0)	C	15
10	2	80 (±15.49)	B	0 (±0)	C	15
	4	98.67 (±3.49)	A	4.67 (±6.94) × 10^3^	BC	15
	6	99.33 (±2.49)	A	5.99 (±9.29) × 10^5^	B	15
25	2	99.33 (±2.49)	A	2.41 (±1.59) × 10^7^	A	15
	4	100 (±0)	A	3.74 (±1.39) × 10^7^	A	15
	6	99.33 (±2.49)	A	3.86 (±1.03) × 10^7^	A	15
35	2	0 (±0)	C	0 (±0)	C	15
	4	0 (±0)	C	0 (±0)	C	15
	6	0 (±0)	C	0 (±0)	C	15

* Mean ± standard deviation. ** Means of one growth parameter with the same letters are not significantly different. (Wilcoxon test, *p* < 0.05).

**Table 4 jof-09-00668-t004:** Effects of the incubation conditions on the conidial size of the *Metarhizium brunneum* strain JKI-BI-1450 on the granules.

Conditions		Conidial Size						
		Conidial Length [µm]		Conidial Width [µm]		Conidial Index *		*n*
Incubation temperature [°C]	10	7.96 (±1.18) **	A ***	2.75 (±0.38)	A	2.93 (±0.44)	A	210
	25	6.62 (±0.50)	B	2.24 (±0.15)	B	2.97 (±0.30)	A	300
Incubation period [weeks]	4	7.18 (±1.12)	A	2.43 (±0.35)	A	2.97 (±0.37)	A	240
	6	7.16 (±1.03)	A	2.47 (±0.37)	A	2.92 (±0.36)	A	270

* The index is calculated by dividing length by width. ** Mean ± standard deviation. *** Means of one condition with one conidial size parameter and the same letters are not significantly different. (GLMM, *p* < 0.05).

**Table 5 jof-09-00668-t005:** Effects of the incubation conditions on the conidial size of the *Metarhizium brunneum* strain JKI-BI-1450 on the granules.

Condition		Conidial Size						
		Conidial Length [µm]		Conidial Width [µm]		Conidial Index *		*n*
Incubation temperature [°C]	Soil temp.	6.96 (±0.54) **	A ***	2.23 (±0.22)	A	3.15 (±0.40)	A	1500
	25	6.38 (±0.55)	B	2.21 (±0.19)	B	2.91 (±0.38)	B	1500
Incubation period [weeks]	2	6.70 (±0.63)	AB	2.19 (±0.20)	B	3.07 (±0.39)	A	600
	3	6.73 (±0.64)	A	2.22 (±0.21)	AB	3.07 (±0.43)	A	600
	4	6.63 (±0.59)	B	2.23 (±0.22)	A	3.00 (±0.43)	B	600
	5	6.66 (±0.63)	AB	2.24 (±0.21)	A	3.00 (±0.42)	B	600
	6	6.63 (±0.59)	B	2.23 (±0.19)	AB	3.00 (±0.38)	B	600

* The index is calculated by dividing the length by the width. ** Mean ± standard deviation. *** Means of one condition at one conidial size parameter with the same letters are not significantly different. (GLMM, *p* < 0.05).

**Table 6 jof-09-00668-t006:** Effects of the incubation temperature on the conidial size of the *Metarhizium brunneum* strain JKI-BI-1450 on MPA dishes after 21 days.

Condition		Conidial Size						
		Conidial Length [µm]		Conidial Width [µm]		Conidial Index *		*n*
Incubation temperature [°C]	15	7.04 (±0.36) **	A ***	2.26 (±0.15)	A	3.13 (±0.26)	A	300
	20	6.97 (±0.37)	A	2.27 (±0.19)	A	3.09 (±0.30)	AB	300
	25	6.47 (±0.51)	C	2.26 (±0.15)	A	2.87 (±0.29)	C	300
	30	6.82 (±0.68)	B	2.24 (±0.17)	A	3.06 (±0.33)	B	300

* The index is calculated by dividing the length by the width. ** Mean ± standard deviation. *** Means of one condition with one conidial size parameter and the same letters are not significantly different. (GLMM, *p* < 0.05).

**Table 7 jof-09-00668-t007:** Effects of the production and incubation temperature of the germination of JKI-BI-1450 after 96 h, and of the germination process over 24 h. τ is the time point where 50% of the maximal germinated conidia are germinated.

Conditions		Germination after 96 h [%]	τ		Slope Inflection		*n*
Production temperature [°C]	15	99.53 (±0.70) *	15.42 (±2.44)	C **	13.69 (±4.65)	AB **	9
	20	99.00 (±1.12)	15.32 (±2.15)	C	15.33 (±5.95)	B	9
	25	99.67 (±0.59)	12.81 (±2.14)	B	13.00 (±2.27)	A	9
	30	99.17 (±1.08)	12.31 (±1.70)	A	14.99 (±3.51)	AB	9
Incubation temperature [°C]	15	99.47 (±0.70)	16.18 (±1.83)	D	12.07 (±4.48)	A	9
	20	99.19 (±1.04)	15.71 (±1.64)	C	11.32 (±2.25)	A	9
	25	99.39 (±0.96)	12.57 (±1.18)	B	15.99 (±3.25)	B	9
	30	99.31 (±1.01)	11.40 (±1.39)	A	17.63 (±3.74)	B	9

* Mean ± standard deviation. ** Means of τ and slope inflection at one condition with the same letters are not significantly different. (GLMM, *p* < 0.05).

**Table 8 jof-09-00668-t008:** Effects of the production and incubation temperature on the virulence of JKI-BI-1450 against *Galleria mellonella*.

Conditions		ST_50_ [d]		Final Mortality [%]		*n*
Control	Water	10.83 (±2.85) *	C **	18.75 (±19.28)	C	16
	0.5% Tween 80^®^	7.63 (±5.16)	B	55.00 (±21.6)	B	16
Production temperature [°C]	15	3.97 (±1.39)	A	98.13 (±5.44)	A	16
	20	4.03 (±1.35)	A	100 (±0)	A	16
	25	4.54 (±1.84)	A	95.63 (±10.31)	A	16
	30	4.04 (±1.74)	A	98.13 (±7.5)	A	16
Incubation temperature [°C]	15	6.03 (±1.31)	C	99.38(±2.5)	A	16
	20	4.06 (±1.32)	B	100 (±0)	A	16
	25	3.33 (±0.76)	AB	100 (±0)	A	16
	30	3.16 (±0.94)	A	92.5 (±12.38)	B	16

* Mean ± standard deviation. ** Means of ST_50_ and final mortality at one condition with the same letters are not significantly different. (GLMM, *p* < 0.05).

## Data Availability

The data that support the findings of this study are available from the corresponding author upon reasonable request.

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
