# Peer review of "Investigation on the Influence of Production and Incubation Temperature on the Growth, Virulence, Germination, and Conidial Size of Metarhizium brunneum for Granule Development"

_jof, 2023, doi:10.3390/jof9060668_

Round 1

Reviewer 1 Report

I consider that the present manuscript has solid elements, but I think that the following information need to be add for the readers and gives more solidity to the work.

• I believe that need talk about the advantages or disadvantages of using granules vs. conidia.

• Talk about humidity % and its effect in the introduction, materials and methods, results and discussion.

• Images of the granules and the size of the conidia need to be add.

Reviewer 2 Report

There are grammatical corrections that should be addressed.  For instance, rather than the noun "conidia", the adjective "conidial" is more appropriate (see page 1, line 20 ...for conidiation 20-25 C.  Conidial size...  )

Change conidiation (p.1, line 37) to conidiate

Drop However, p. 1. line12, just start with "We investigated...

Instead of proved, use demonstrated (p. 1, line 43); same on P.2. line 83, not proofed, but tested

P. 2, l 45, Only M. robertsii grew

Consider less us of however, thereafter, etc., for instance, P. 3, l 109, New MP plates were then inoculated with the fungus produced at these temperatures and incubated fro 14 days at the same temperatures as before.

I am suggesting that you can eliminate extra words in several of the sentences.  P. 3, lines1 17-120  For liquid production of fungal biomass, 50 mls of medium containing 2.5% ... sodium chloride (Merck)  in 100-ml flasks was used.

Are the conditions for the fluid-bed dryer not described in a previous paper?  If so cite the paper and eliminate lines 136-140.  Lines 147-150 could be shortened.

P. 4, lined 153---just one name, "conidial index" , especially as this name is used in Table 2

Page 4, line 168; what is being presented in Fig 2 for soil temperature?  Are the wavy lines in Fig. 2 four hr averages?

Page 4, lines 170-172.  If you sampled every week for 6 weeks that is 6 samples.  You state 30 petri dishes were inoculated, but you sampled 10 dishes for each week.  This would seem you needed 60 dishes for each treatment or do you mean you sampled 5 petri dishes for each treatment?

Page 4 section 2.8.1---too many words; eliminate deep freezer, inoculation loop (you made a conidial suspension in 0.5% Tween in a 1.5 ml tube, votrexed for 10 sec, and and incubated in a sonic bath for 15 minutes.  Conidial concentration was determined with a haemocytometer, and a conidial suspension of 1 x 10 6 was made for each treatment.)

P.2, l 61, enzymatic instead of enzymes

Section 2.8.3---is this germination rate or just final germination percentage

Section 2.8.4---too many words, reduce excess, such as deep freezer, sterile spatula, etc.  How was mortality determined?

Section 2.9---Kruskal-Wallis and Wilcoxon test are both non-parametric tests.  Explain the reason both are used as described in this section.

How was the tau parameter determined from the two equations presented in section 2.9

Section 3.1---instead of and use or--25 or 30 C; eliminate moreover, instead  JKI-BI-1450 did not grow or sporulate at 37 C.

Section 3.2---line 277 suggest ...hundred times lower than the other granules and differed significantly form granules at other production temperatures.  Text (line 280) says 3.16, but table says 3.17---should be the same

line 292  either ..., the conidial size OR ..., the size of conidia were

Section 3.4--I am lost on the explanation of these results; it would be helpful if the authors would again indicate what is being compared (soil vs 25 C at each sampling period, soil or 25 C across all sampling periods, or some other combination); the authors state at week 2 and 6 there are significant differences between soil and 25C, but there are no differences between weeks 2,3,4 or 5 according to the small case letters for 25C ; for soil the only difference seems to be week 1; I am confused, please explain.  I also do not understand the analysis for conidia per granule.

line 370 should be ...at simulated soil temperatures

Section 3.5--how was the inflection point determined for each line presented in Fig. 3?  On page 5, formula 2, which is used to determine the slope at the inflection point, does one define the inflection point for each production temperature?  Also not all the variables are defined for formula 2 (what is q and what is a design parameter.  To calculate tau, one would have to define P and Pmax (what is the difference between these two variables).

Page 14, lines 481-482; the issue is that you have no data on any physiological parameter of your agent to support this statement.

P. 14, line 487-490 I suggest that you examine the biomass microscopically to determine if this statement is valid (one should see large differences in spore vs mycelia if this is true, a very simple experiment)

P 14, lines 501-506 does the presence of bacterial contamination affect your overall conclusions (were granules that were contaminated at higher production temperatures affect results of say mortality experiments)

P 15, lines 518-521; this should be expected as the simulated soil temperatures fluctuated greatly on any given day as well as over the period of the experiment; these data are more important than the laboratory experiments at constant temperatures.

Line 540---"proves", I would say strongly suggests as you have not examined but only a few fungi

Line 556--I do not think this is germination rate, but simply final germination percentage

Line 557  do you have evidence that insects strip off condia;

Lines 547-568 contain extraneous words; this paragraph should be shorter; for instance delete line 559; the reference to Rath is not very relevant

Page 16, lines 592  There are other simple tests that could be attempted prior to field application.  Do the granules perform in a soil simulated temperature against a target pathogen, for instance.

I would suggest a summary paragraph at the end of the discussion to repeat the most salient features of the manuscript.

Abstract: see suggestions above; line 23  delete positive

I\

see above comments
